# Factors Associated with Anxiety, Depression, and Quality of Life in Patients with Psoriasis: A Cross-Sectional Study

**DOI:** 10.3390/brainsci14090865

**Published:** 2024-08-27

**Authors:** Salvatore Cipolla, Pierluigi Catapano, Antonio Fiorino Bonamico, Valeria De Santis, Roberta Murolo, Francesca Romano, Antonio Volpicelli, Francesco Perris, Ada Lo Schiavo, Michele Fabrazzo, Francesco Catapano

**Affiliations:** 1Department of Psychiatry, University of Campania “Luigi Vanvitelli”, Largo Madonna delle Grazie 1, 80138 Naples, Italy; pierluigi.catapano@unicampania.it (P.C.); antoniofiorino.bonamico@studenti.unicampania.it (A.F.B.); valeria.desantis@unicampania.it (V.D.S.); roberta.murolo@unicampania.it (R.M.); antonio.volpicelli@policliniconapoli.it (A.V.); francesco.perris@unicampania.it (F.P.); michele.fabrazzo@unicampania.it (M.F.); francesco.catapano@unicampania.it (F.C.); 2Dermatology Unit, University of Campania “Luigi Vanvitelli”, Via S. Pansini 5, 80131 Naples, Italy; dott.francescaromano@gmail.com (F.R.); adaloschiavo54@gmail.com (A.L.S.)

**Keywords:** psoriasis, quality of life, anxiety, depression, affective disorders, mental health, psychiatric comorbidities

## Abstract

Background: Psoriasis is a chronic skin disorder affecting 2–3% of the global population, and is associated with several comorbidities, including psychiatric disorders. This study aimed to identify factors influencing anxiety, depression, and quality of life (QoL) in patients with psoriasis. Methods: This observational study included 112 patients diagnosed with psoriasis. Dermatological and psychiatric assessments were conducted using Psodisk, the Hamilton Anxiety Rating Scale (HAM-A), Hamilton Depression Rating Scale (HAM-D), Symptom Checklist-90-Revised (SCL-90-R), and 36-Item Short Form Health Survey (SF-36). Descriptive statistics, correlation analyses, and multivariate regression models were employed. Results: The sample was predominantly middle-aged males (mean age 48.91 years). Females (*p* < 0.001), patients with arthritis (*p* < 0.05), and those with a sedentary lifestyle (*p* < 0.05) showed higher anxiety and depression scores. Psodisk subscales significantly correlated with psychiatric symptoms and QoL measures (*p* < 0.001). Pain (B: 0.63, *p* < 0.05; B: −2.03, *p* < 0.01) and sleep disturbances (B: 0.68, *p* < 0.01; B: 0.60, *p* < 0.01; B: −1.46, *p* < 0.01; B: −1.57, *p* < 0.05; B: 3.91, *p* < 0.05) emerged as major predictors of poor mental health and reduced QoL. Conclusions. The study underscores the complex relationship between psoriasis, psychiatric comorbidities, and QoL. Key factors exacerbating anxiety and depression include female gender, arthritis, and sedentary lifestyle. Comprehensive management of psoriasis should address both dermatological and psychological aspects, with a focus on pain relief and improving sleep quality to enhance overall patient well-being.

## 1. Introduction

Psoriasis is a chronic, immune-mediated skin disorder characterized by the presence of erythematous and scaly plaques, affecting about 2–3% of the global population [1]. The etiopathogenesis of psoriasis is complex and still partially unknown: it seems to involve a complex interplay between genetic predisposition and environmental factors, and those risk factors may act together as triggers, leading to an aberrant immune response [2,3,4,5]. This results in the hyperproliferation of keratinocytes and chronic inflammation [6].

Psoriasis can be associated with systemic comorbidities, including cardiovascular diseases, hypertension, metabolic syndrome, obstructive sleep apnea, other inflammatory diseases [7,8], and psychiatric disorders [9,10,11,12]. A systematic review and meta-analysis reported that the prevalence of depression in patients with psoriasis ranges from 20% to 30%, while the prevalence of anxiety ranges from 15% to 20% [13]. Furthermore, higher rates of self-harm and suicidality have been found in patients with psoriasis when compared with the general population or individuals with different dermatological conditions [14,15,16], as well as sleep disorders and sexual dysfunctions [17,18,19,20].

The psychological burden of psoriasis is substantial and multifaceted. Patients frequently experience stigma, social isolation, and reduced self-esteem due to the visible and often disfiguring nature of the disease [21,22]. These psychosocial stressors can precipitate or exacerbate psychiatric conditions such as anxiety and depression [23,24,25]. The psychological and psychiatric morbidity is often a significant proxy of distress, even greater than that associated with the dermatological manifestations of the disorder [26].

However, the relationship between dermatological and psychiatric symptoms is still unclear [27], being complex to ascertain to which extent psychiatric symptoms are due to the underlying psoriasis rather than to the psychosocial implications derived from living with a chronic disease [28]. In fact, psychiatric comorbidities may be intertwined with the pathophysiology of psoriasis itself, as the chronic inflammatory state inherent to psoriasis has been implicated in the pathogenesis of many psychiatric disorders such as anxiety and depression [29]. This overlap underscores the importance of addressing both physical and mental health aspects in the management of patients with psoriasis. Furthermore, the presence of psychiatric comorbidities can adversely affect treatment adherence [30,31,32], disease severity [13,33], and overall prognosis, thereby compounding the disease burden [15,23].

It is not surprising that perceived stress in patients with psoriasis, as well as in several other chronic inflammatory diseases, may predict a poor quality of life (QoL) [34]. Studies have consistently shown that patients with psoriasis score lower on QoL measures compared to healthy controls, with the severity of skin lesions and the presence of arthritis being significant contributors [35,36,37,38]. Quality of life is a critical consideration in the holistic management of psoriasis, and the impact on QoL is not limited to physical discomfort, but extends to mental and emotional well-being and the resulting social and work disability [39].

Given the intricate relationship between psoriasis, psychiatric disorders, and QoL, it is essential to identify factors that influence these outcomes [40,41]. This knowledge can inform targeted interventions aimed at improving overall patient well-being [42]. However, comprehensive analyses incorporating multiple variables and their interactions are limited. While individual studies have examined the impact of specific factors like disease severity [43], treatment adherence [44,45], and psychiatric comorbidities [40] on QoL, few of them have integrated these elements into a coherent model that can guide comprehensive patient care strategies. Recent research has started to unravel these complex interactions. Despite this, gaps remain in understanding the multifactorial influences on QoL in patients with psoriasis, particularly regarding the interplay between physical symptoms, mental health, and lifestyle factors such as physical activity and sleep quality.

In this study, we aimed to identify factors influencing anxiety, depression, and the deterioration in mental and physical QoL in patients with psoriasis. Using a comprehensive analytical approach, we aimed to clarify the interactions among multiple variables, including demographic characteristics, lifestyle factors, disease severity, and psychiatric comorbidities. Understanding these relationships is crucial for developing comprehensive, patient-centered management strategies that address both the dermatological and psychological aspects of the disease. Specifically, we focused on the roles of pain and sleep disturbances in predicting mental health outcomes and overall QoL, thereby highlighting potential targets for therapeutic interventions.

## 2. Materials and Methods

### 2.1. Study Design and Settings

This was a cross-sectional observation study carried out in a real-world setting. The recruitment of patients took place in the outpatient unit of the Department of Dermatology of the University of Campania Luigi Vanvitelli. Patients were included if they met the following criteria: (1) age between 18 and 70 years; (2) diagnosis of psoriasis by an experienced dermatologist, according to NICE guidelines [46]; (3) ongoing topical and/or systemic treatment for psoriasis; and (4) willingness to undertake a psychiatric assessment. Exclusion criteria were: (1) inability to give written consent to participate in the study; (2) lack of proficiency in the Italian language; (3) presence of other physical diseases not stabilized or untreated at the time of enrollment; and (4) personal history of alcohol or substance abuse.

All patients gave their written informed consent to participate in the study after receiving a full description of the study aims and design. The study was carried out in accordance with the latest version of the Declaration of Helsinki for experiments involving humans and was approved by the Ethics Committee of the University of Campania Luigi Vanvitelli (130/2014).

### 2.2. Assessment Tools

Patients’ sociodemographic and clinical characteristics, including gender, age, marital and employment status, educational level, lifestyle habits, age of psoriasis onset, previous and ongoing treatments, and family and personal history of psychiatric disorders were recorded with an ad hoc schedule at the time of recruitment.

Dermatological assessment was performed by an experienced dermatologist. Psychopathological status of each enrolled patient was assessed by an experienced psychiatrist. Only validated Italian versions of assessment instruments were adopted.

#### 2.2.1. Psodisk

Each dermatological assessment included the administration of Psodisk, a user-friendly 10-item questionnaire designed to capture the multifaceted nature of psoriasis providing insights into how psoriasis affects various aspects of daily living [47,48]. Psodisk is meant to be completed by both patients and physician together, and the answers are given on a 10-point visual analogue scale, ranging from “absolutely not” to “definitely yes,” graphically represented on a disk. The tool consists of ten subscales, each targeting a different domain of life affected by psoriasis: health, physical pain, itch, sleep, social life, work and other daily activities, peace of mind, sexual life, shame, and skin involvement. Responses are typically rated on a Likert scale, with higher scores indicating a greater negative impact of psoriasis. Psodisk was used as covariate, and Cronbach’s alpha for the Italian version of Psodisk is 0.927 [48].

#### 2.2.2. Hamilton Anxiety Rating Scale (HAM-A)

The Hamilton Anxiety Rating Scale (HAM-A) is a 14-item clinician-related questionnaire. Each item is scored on a 5-point scale, and a higher total score indicates more anxiety symptoms [49]. HAM-A total score was used as outcome variable to assess the level of anxiety in participants, and for the Italian version of the scale, Cronbach’s alpha was found to be 0.86 [50].

#### 2.2.3. Hamilton Rating Scale for Depression (HAM-D)

The Hamilton Rating Scale for Depression (HAM-D) is a 21-item questionnaire assessing depressive symptoms, with higher scores indicating severe depression symptoms [51]. The HAM-D served as outcome variable to measure depressive symptoms, and the Italian version of this tool has a Cronbach alpha of 0.833 [52].

#### 2.2.4. Symptom Checklist-90-Revised (SCL-90-R)

The Symptom Checklist-90-Revised (SCL-90-R) is a self-report inventory designed to evaluate a broad range of psychological problems and psychopathological symptoms that consists of 90 items, each rated on a 5-point scale of distress (from “not at all” to “extremely”). These items are organized into nine primary symptom dimensions: (1) somatization (SOM), (2) obsessive–compulsive (O–C), (3) interpersonal sensitivity (I-S), (4) depression (DEP), (5) anxiety (ANX), (6) hostility (HOS), (7) phobic anxiety (PHOB), (8) paranoid ideation (PAR), and (9) psychoticism (PSY). The Global Severity Index (GSI) reflects the overall level of psychological distress, and higher scores indicate greater psychological distress and symptom severity [53,54]. The GSI was used as outcome variable in statistical analyses, and the Italian version of SCL-90-R has a Cronbach alpha of 0.96 [55].

#### 2.2.5. 36-Item Short Form Health Survey (SF-36)

The health-related quality of life was assessed by the 36-Item Short Form Health Survey (SF-36), a widely used questionnaire designed to capture a broad range of health dimensions and applicable to diverse patient populations, making it a valuable tool in both clinical practice and research. The SF-36 consists of 36 items that cover eight health domains, divided into two main categories: physical and mental health. The physical component summary (PCS) aggregates scores from the following four physical health domains: (1) physical functioning (PF), (2) role limitations due to physical health (RP), (3) bodily pain (BP), and (4) general health (GH). The mental component summary (MCS) aggregates scores from the following four mental health domains: (1) vitality (VT) or energy/fatigue, (2) social functioning (SF), (3) role limitations due to emotional problems (RE), and (4) mental health (MH) or emotional well-being [56]. Higher scores refer to a better health-related quality of life. PCS and MCS scores were used as outcome variables, and the Italian version of the SF-36 has a Cronbach alpha > 0.77 [57].

### 2.3. Statistical Analysis

The statistical data related to the sociodemographic and clinical characteristics of the sample, as well as those from other relevant assessment instruments, were calculated and are presented as means and standard deviations (SDs) or frequencies and percentages (%), as appropriate. The Kolmogorov–Smirnov test was adopted to check the normality of distribution of our sample. Student’s *t*-test for independent samples was performed to assess differences in HAM-A, HAM-D, PCS, MCS and GSI mean scores with discrete variables. Correlation analyses were performed to explore the association between those same scores and continuous variables, including Psodisk subscales. A generalized linear regression model was finally applied to detect possible predictors of anxiety, depression, psychopathological distress, and physical and mental quality of life in our cohort of patients. The level of statistical significance was set at *p* < 0.05. The data were analyzed using IBM Statistical Package for Social Sciences (SPSS), version 26.

## 3. Results

### 3.1. Sociodemographic Characteristics of the Sample

The total sample consisted of 112 patients, mainly male (67, 59.8%) and with a mean age of 48.91 ± 12.73 years. Almost half of them were middle school graduates (41.1%) and only 7.1% of the sample had obtained a university degree. Mean body mass index (BMI) was 29.33 ± 6.45 and 89 subjects (79.5%) declared having a sedentary lifestyle, with low-to-absent physical activities performed during the day. Slightly more than half of the subjects (54.5%) were smokers and 97.3% drank alcohol regularly. Nevertheless, these data fall within regular alcohol consumption in the context of the Mediterranean diet [58] and are not indicative of an alcohol use disorder, as this was an exclusion criterion. Most of the sample (64.3%) reported another dermatological disease. Our cohort included patients diagnosed with psoriasis with a median illness duration of 17.4 ± 12.49 years. In addition, in 31 cases (27.7%), arthropathic psoriasis was diagnosed. Mean Psodisk total score was 52.24 ± 26.22. Overall, 14% of the sample had a positive family history for psychiatric disorders, nine subjects (8%) had a personal history of psychiatric disorders, twelve patients (10.7%) were in treatment with anxiolytics, and nine (8%) were on antidepressant drugs. No other psychiatric medications (such as antipsychotics and mood stabilizers) were mentioned by the enrolled subjects. Mean HAM-A total score was 12.59 ± 9.56, and mean HAM-D total score was 11.93 ± 7.67. Mean GSI scored 60.79 ± 53.63. The main sociodemographic and clinical characteristics of the sample are reported in Table 1 and Table 2, respectively.

### 3.2. Univariate Analyses

To assess differences in psychopathological scores within discrete variables, independent samples *t*-tests were performed (Figure 1). The mean HAM-A total score was statistically significantly higher in female subjects (d = 7.15; *t* = 4.16; *p* < 0.001), in patients who reported a positive personal history of psychiatric disorders (d = 7.57; *t* = 2.32; *p* = 0.02), and in patients with arthritis (d = 5.39; *t* = 2.48; *p* = 0.02). Furthermore, the mean HAM-D total score was higher in female patients (d = 5.39; t = 3.87; *p* < 0.001) and in patients with a sedentary lifestyle (d = 4.01; *t* = 2.68; *p* = 0.005). The mean PCS score was significantly lower in females (d = 11.02; *t* = 2.45; *p* = 0.016) and in patients with arthritis (d = 11.87; *t* = 2.40; *p* = 0.018), indicating a greater impairment of the physical component of quality of life recorded with the SF-36. On the other hand, the mean MCS score was significantly lower in female patients (d = 13.22; *t* = 3.17; *p* = 0.002. Finally, female patients showed a significantly higher score than male patients on the GCI scale (d = 33.11; *t* = 3.35; *p* = 0.001). No statistically significant differences were found in the scores of the psychopathological scales considered in the patient groups defined on the basis of smoking habits, alcohol consumption, or family history of psychiatric disorders.

In Table 3, results of the correlation analyses between age, BMI, duration of psoriatic illness, Psodisk subscales, and psychopathological variables such as HAM-A, HAM-D, PCS, MCS ,and GSI scores are reported. The analysis evidenced that age and BMI were inversely correlated with PCS (r = −0.242, *p* = 0.010; r = −0.222, *p* = 0.018 respectively) and MCS (r = −0.204, *p* = 0.031; r = −0.225, *p* = 0.017 respectively) subscales of the SF-36. No significant correlation was found between duration of psoriatic illness and the psychopathological scales. Significant correlations were also found between most Psodisk subscales and the psychopathological variables considered, with the only exception of the work and other physical activities dimension and the HAM-D total score. In particular, health subscales of Psodisk correlated with HAM-A and HAM-D total scores (r = 0.500, *p* < 0.001 and r = 0.462, *p* < 0.001, respectively) and with GSI, PCS and MCS scores (r = 0.493, *p* < 0.001; r = −0.523, *p* < 0.001; r = −0.464, *p* < 0.001, respectively); the pain subscale of Psodisk also correlated with HAM-A and HAM-D total scores (r = 0.525, *p* < 0.001; r = 0.415, *p* < 0.001) and with GSI, PCS and MCS scores (r = 0.442, *p* < 0.001; r = −0.563, *p* < 0.001; r = −0.443, *p* < 0.001). The itching item of Psodisk correlated with HAMA-A (r = 0.321, *p* = 0.001), HAM-D (r = 0.313, *p* = 0.001), GSI (r = 0.359, *p* < 0.001), PCS (r = −0.278, *p* = 0.003) and MCS (r = −0.389, *p* < 0.001). Psodisk sleep subscale correlated with HAM-A (r = 0.456, *p* < 0.001), HAM-D (r = 0.409, *p* < 0.001), GSI (r = 0.467, *p* < 0.001), PCS (r = −0.435, *p* < 0.001), and MCS (r = −0.488, *p* < 0.001). Social life subscale correlated with HAM-A and HAM-D (r = 0.331, *p* < 0.001; r = 0.220, *p* = 0.020), and with GSI, PCS and MCS (r = 0.383, *p* < 0.001; r = −0.228, *p* = 0.016; r = −0.324, *p* = 0.001). Work and other daily activities Psodisk item correlated with HAM-A (r = 0.265, *p* = 0.005), GSI (r = 0.246, *p* = 0.009), PCS (r = −0.345, *p* < 0.001), and MCS (r = −0.349, *p* < 0.001). The Psodisk subscale of peace of mind correlated with HAM-A (r = 0.399, *p* < 0.001), HAM-D (r = 0.333, *p* < 0.001), CGI (r = 0.405, *p* < 0.001), PCS (r = −0.390, *p* < 0.001), and MCS (r = −0.520, *p* < 0.001). The sexual life item of Psodisk showed correlations with HAM-A (r = 0.241, *p* = 0.011), HAM-D (r = 0.197, *p* = 0.038), GSI (r = 0.356, *p* < 0.001), PCS (r = −0.245, *p* = 0.010), and MCS (r = −0.320, *p* = 0.001). The shame item correlated with HAM-A (r = 0.331, *p* < 0.001), HAM-D (r = 0.289, *p* = 0.002), GSI (r = 0.407, *p* < 0.001), PCS (r = −0.237, *p* = 0.012), and MCS (r = −0.346, *p* < 0.001). Finally, the skin involvement measured by Psodisk showed correlations with HAM-A and HAM-D (r = 0.288, *p* = 0.002; r = 0.315, *p* = 0.001), and GSI, MCS, and PCS (r = 0.341, *p* < 0.001; r = −0.224, *p* = 0.018; r = −0.294, *p* = 0.002).

### 3.3. Multivariate Analysis

The results of the regression linear analysis are reported in Table 4. The model was run to assess the independent predictors associated with the psychopathological scales.

Patients with higher HAM-A scores were more likely to be female (B = 3.92, *p* = 0.026), have a shorter duration of illness (B = −0.14, *p* = 0.026), and experience more physical pain (B = 0.62, *p* = 0.034) and sleeping disturbance (B = 0.68, *p* = 0.007) due to the psoriasis. Higher HAM-D scores were more strongly associated with being female (B = 3.42, *p* = 0.031), leading a sedentary lifestyle (B = 4.02, *p* = 0.022), and experiencing sleep disturbance caused by psoriasis as evaluated by the Psodisk subscale (B = 0.60, *p* = 0.008). A worse PCS score was associated with the presence of pain and sleep disturbance due to psoriasis (B = −2.03, *p* = 0.008; B = −1.46, *p* = 0.025), whereas a worse MCS score was linked to sleep disturbance (B = –1.57, *p* = 0.010) and Psodisk subscale peace of mind (B = −2.30, *p* = 0.006). Additionally, sleep disturbance also correlated with higher GSI scores (B = 3.91, *p* = 0.012).

## 4. Discussion

This study provides valuable insights into the sociodemographic characteristics, lifestyle factors, and psychological comorbidities of patients with psoriasis. Notably, a significant portion of the sample led a sedentary lifestyle (79.5%), and a considerable percentage were smokers (54.5%) and regular alcohol consumers (97.3%). These lifestyle factors may contribute to the overall health burden and quality of life in patients with psoriasis [59,60]. The prevalence of psychiatric comorbidities in our cohort was noteworthy, with 8% of patients having a personal history of psychiatric disorders and 10.7% receiving anxiolytic treatment at the time of recruitment. This aligns with previous studies highlighting the high prevalence of psychological issues in dermatological conditions like psoriasis [23,33]. Patients with psoriasis are frequently burdened with mental health issues due to the chronic and visible nature of their condition, which can significantly affect their social interactions and self-esteem [61]. Furthermore, female patients exhibited higher levels of anxiety (assessed by HAM-A) and depression (assessed by HAM-D) compared to males, and those with a sedentary lifestyle showed significantly higher depression scores. These findings are consistent with existing literature that indicates gender and physical activity levels as critical determinants of mental health in chronic illness [18,61,62,63,64,65]. The higher prevalence of anxiety and depression in female patients may be attributed to both biological and social factors, including hormonal fluctuations and societal pressures [66,67]. An important biological factor to consider is the role of inflammation, which is a common denominator in both psoriasis and psychiatric disorders. Psoriasis is characterized by chronic systemic inflammation, which can affect various organs, including the brain [68,69]. Pro-inflammatory cytokines such as tumor necrosis factor-alpha (TNF-α), interleukin 6 (IL-6), and interleukin 1 beta (IL-1β) are elevated in psoriasis and have been implicated in the pathophysiology of depression and anxiety [70,71]. These cytokines can cross the blood–brain barrier and influence neurotransmitter systems, leading to changes in mood and behavior. Furthermore, chronic inflammation can alter the hypothalamic–pituitary–adrenal (HPA) axis, leading to increased levels of cortisol, which is associated with both psoriasis severity and depressive symptoms [72]. Finally, an increase in inflammation can also result from unhealthy lifestyles such as physical inactivity, resulting in a complex interplay that may further contribute to the mental health challenges faced by patients with psoriasis [73]. It should be noted that treatment of psoriasis may involve biologics that affect blood cytokine levels (i.e., IL-12, IL-23, and TNF-α), with an overall reduction in inflammation [74]. It is conceivable that these therapies may also impact anxiety and depressive symptom severity, thereby potentially improving patients’ mental health and well-being. However, this hypothesis is merely speculative, considering that the impact of biological therapies on psychiatric symptoms was not the objective of this study. Nonetheless, this could be the starting point for further studies, which should adopt a longitudinal design and should be carried out on larger samples.

Our correlation analysis indicated that both age and BMI negatively correlated with the physical and mental components of quality of life (PCS and MCS scores of SF-36), suggesting that older and overweight patients may experience greater impairments in both the physical and mental components of QoL. These findings underscore the importance of managing comorbid conditions such as obesity in patients with psoriasis, as these can exacerbate the overall disease burden [75]. Additionally, Psodisk subscales correlated significantly with psychopathological variables, reinforcing the impact of disease severity on mental health. This correlation emphasizes the need for comprehensive assessments that include both physical and psychological aspects of the psoriasis.

The multivariate analysis further substantiated these findings by confirming the correlations among gender and sedentary lifestyle behaviors and levels of anxiety and depressive symptoms and by identifying other independent variables associated with psychopathological outcomes. According to these analyses, disease duration of psoriasis was negatively associated with anxiety symptoms, indicating that longer illness duration might contribute to better coping mechanisms or adaptation over time. This adaptive response may be due to patients developing more effective strategies for managing their condition and its psychosocial impact over time [76]. However, this interpretation is merely speculative and requires further investigations to be confirmed.

Physical pain and sleep disturbances, as assessed by Psodisk, were significantly associated with both anxiety and depressive symptoms, as well as lower quality-of-life scores. This highlights the critical role of pain management and sleep quality in improving the overall well-being of patients diagnosed with psoriasis [77]. Pain is a common symptom in psoriasis that can severely limit daily activities and negatively impact mental health. Similarly, sleep disturbances are frequently reported in patients with psoriasis and are associated with higher levels of anxiety and depression [15]. In fact, previous research has shown that the prevalence of sleep disorders in these patients is significantly higher than in the general population, with conditions such as insomnia, obstructive sleep apnea, and restless legs syndrome being particularly common [77,78,79]. These sleep disorders not only exacerbate the physical symptoms of psoriasis but also contribute to a vicious cycle of psychological distress [80]. Once again, the association between sleep disturbances and mental health issues in psoriasis can be partly explained by the role of systemic inflammation [77]. Pro-inflammatory cytokines, which are elevated in psoriasis, can disrupt sleep architecture, leading to poor sleep quality and increased daytime fatigue [81]. This disruption in sleep can further elevate inflammatory markers, creating a feedback loop that exacerbates dermatological and psychiatric symptoms.

Moreover, the burden of psoriasis, including the stress of managing a chronic and visible skin condition, can contribute to sleep disorders: anxiety about flare-ups, social stigma, and self-esteem issues can lead to hyperarousal, making it difficult for patients to fall and stay asleep. The chronic sleep deprivation that results can impair cognitive function, mood regulation, and overall mental health, further highlighting the importance of addressing sleep disturbances in this population. Addressing these symptoms through targeted interventions can substantially improve patient outcomes. In particular, cognitive–behavioral therapy for insomnia (CBT-I) has shown promise in improving sleep and reducing symptoms of depression and anxiety in various populations, and could be particularly beneficial for patients with psoriasis [82,83,84].

Furthermore, we found a significant association between the Psodisk item “peace of mind” and the MCS score of the SF-36 scale, which highlights the profound impact of psoriasis on patients’ psychological stability. The “Peace of mind” Psodisk subscale assesses the psychological tranquility and emotional stability of patients, reflecting their overall mental state, encompassing aspects such as emotional well-being, role limitations due to emotional problems, and social functioning. Interventions aimed at enhancing patients’ peace of mind, such as mindfulness-based therapies and stress management programs, could potentially improve the overall mental health and quality of life in this population [85,86].

Finally, these findings highlight the importance of a multidisciplinary approach in managing psoriasis. Dermatologists, mental health professionals, and primary care providers should collaborate to address the comprehensive needs of patients [87]. Integrating psychological support, lifestyle counseling, and psychiatric assessment into routine dermatological care could potentially mitigate the mental health challenges faced by these patients, leading to more holistic and effective treatment outcomes [18,88,89,90,91,92]. Additionally, the significant impact of sedentary lifestyle on depression scores suggests that promoting physical activity could be a beneficial component of psoriasis management. Regular exercise has been shown to reduce symptoms of depression and anxiety, improve cardiovascular health, and enhance overall quality of life [93,94]. Encouraging patients to engage in physical activities suited to their capabilities can be a practical step towards better mental health outcomes. Furthermore, the lack of significant differences in psychopathological scores based on smoking habits, alcohol consumption, and family history of psychiatric disorders does not undermine the importance of addressing these behaviors, as they still pose significant health risks and can potentially exacerbate other aspects of psoriasis [95,96].

The results of the present paper should be considered in light of several limitations. First, the observational design precludes any conclusions about the causal relationships between considered variables and psychological outcomes. Longitudinal studies are needed to determine the directionality and temporal sequence of these associations. Second, the use of self-reported measures may introduce response bias, as participants might underreport or overreport their symptoms based on their subjective perceptions, and future studies should consider incorporating objective measures. Third, the Hamilton rating scales for depression and anxiety (HAM-D and HAM-A) were originally designed to describe rather than quantify symptoms. Although these instruments have been widely employed for quantitative assessments in clinical research, their use for this purpose may not provide the most accurate representation of symptom severity, considering the availability of more specific and updated assessment instruments. However, given their extensively use in already published articles, the adoption of HAM-D and HAM-A can provide comparable data. Nevertheless, further steps of the present research will include the use of more recent and updated assessment instruments for depressive and anxiety symptoms in order to balance this limitation. Fourth, the relatively limited number of patients and their heterogeneity concerning psoriasis symptoms and clinical subtype limit the generalizability of the findings. Larger samples would enhance the validity of the results. Additionally, the study was conducted in a single geographic region, which may not reflect the experiences of patients with psoriasis in different contexts. Comparative studies across different populations are necessary to confirm the generalizability of these findings. Lastly, another possible limitation of our study is that several other variables may have influenced patients’ anxiety and depressive symptom severity as well as their quality of life. These include the presence of comorbidities, although in their stable phase, any stressful life events that occurred in the months prior to enrollment and the use of psychiatric and dermatological medications. Specifically, the use of anxiolytic or antidepressant medications may have influenced the average HAM-A and HAM-D scores in the overall sample. However, the impact of this confounding factor is likely to be minimal, given the widespread use of these medications [97,98,99]. Similarly, certain dermatological therapies commonly used for the treatment of psoriasis may contribute to the development of affective or anxiety symptoms in patients (i.e., corticosteroids, methotrexate, cyclosporine, and biological therapies). However, we attempted to mitigate this confounding effect by using assessment tools specifically designed for evaluating quality of life and symptoms related to psoriasis, such as Psodisk. Despite these efforts, given the cross-sectional design of the study and its real-world setting, we were not able to fully determine the extent to which these medications influence mental health outcomes. Future research will be essential to explore the implications of these treatments on the mental health of patients with psoriasis more comprehensively.

Despite these limitations, this study provides important insights into the complex interplay between physical and psychological health in individuals with psoriasis and underscores the need for comprehensive care approaches.

## 5. Conclusions

This study underscores the complex and multifaceted nature of psoriasis, which extends well beyond dermatological symptoms to encompass significant psychological, psychiatric, and quality-of-life impairments. Pain and sleep disturbances emerged as key factors influencing both anxiety and depressive symptoms in patients with psoriasis, as well as the overall quality of life. Inflammation appears to be a pivotal factor linking unhealthy lifestyles, psoriasis, psychiatric symptoms such as anxiety and depression, and sleep disturbances.

These insights emphasize the need for a comprehensive approach to psoriasis management that includes addressing psychiatric comorbidities and lifestyle modifications, particularly focusing on improving physical activity, pain management, and sleep quality. Future research should aim to explore targeted interventions that can alleviate these psychic burdens and enhance the quality of life for patients with psoriasis. Integrating psychiatric support and lifestyle counseling into routine dermatological care could potentially mitigate the mental health challenges faced by these patients, leading to more holistic and effective treatment outcomes.

## Figures and Tables

**Figure 1 brainsci-14-00865-f001:**
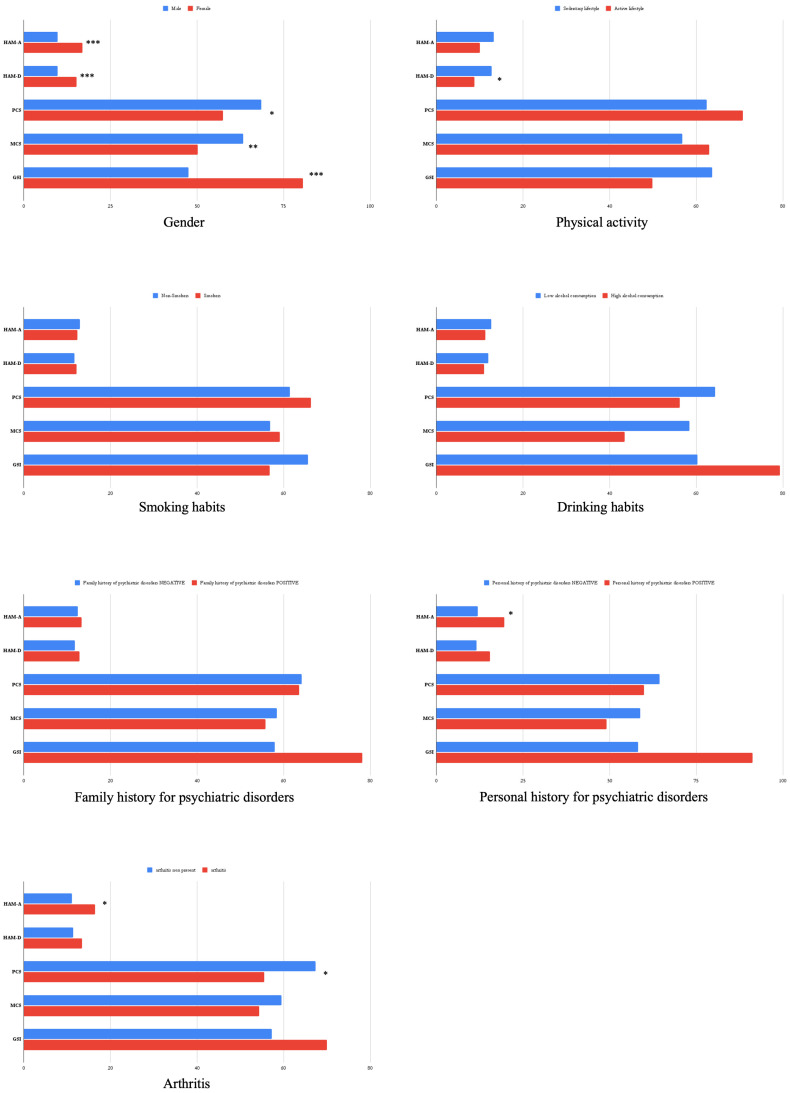
Student’s *t*-tests. Comparison of assessment scale scores by sociodemographic and clinical characteristics in the study population (statistically significant results are highlighted as follows: * *p* < 0.05; ** *p* < 0.01; *** *p* < 0.001).

**Table 1 brainsci-14-00865-t001:** Sociodemographic characteristics of the sample (N = 112).

	*n*	%
Gender		
Male	67	59.8
Female	45	40.2
Age (years)	48.91	±12.73
Education		
Elementary school	26	23.2
Middle school	46	41.1
High school	32	28.6
University	8	7.1
Marital status		
With partner	87	77.7
Without partner	25	22.3
Employed, yes	56	50

Note: Data for continuous variables are expressed as the mean ± standard deviation.

**Table 2 brainsci-14-00865-t002:** Clinical characteristics of the sample (N = 112).

	*n*	%
BMI	29.33	±6.45
Sedentary lifestyle, yes	89	79.5
Smoker, yes	61	54.5
Alcohol consumption	109	97.3
Family history of psychiatric disorders, yes		
Positive	16	14.3
Personal history of psychiatric disorders		
Positive	9	8
Ongoing psychiatric drugs administration		
Anxiolytic	12	10.7
Antidepressant	9	8
HAM-A total score	12.59	±9.56
HAM-D total score	11.93	±7.67
GSI	60.79	±53.63
DoPI (years)	17.4	±12.49
Arthritis	31	27.7
Psodisk total score	52.24	±26.22

BMI: body mass index; DoPI: duration of psoriatic illness; GSI: Global Severity Index; HAM-A: Hamilton Anxiety Rating Scale; HAM-D: Hamilton Rating Scale for Depression. Note: Data for continuous variables are expressed as the mean ± standard deviation.

**Table 3 brainsci-14-00865-t003:** Pearson’s rho correlation between age, BMI, duration of psoriatic illness, Psodisk subscales and psychiatric assessment tool scores (N = 112).

			SF-36		SCL-90
	HAM-A	HAM-D	PCS	MCS	GSI
Age	0.108	0.005	−0.242 *	−0.204 *	0.032
BMI	0.182	0.081	−0.222 *	−0.225 *	0.099
DoPI	−0.089	−0.086	−0.022	0.094	−0.048
Psodisk					
Health	0.500 ***	0.462 ***	−0.523 ***	−0.464 ***	0.493 ***
Pain	0.525 ***	0.415 ***	−0.563 ***	−0.443 ***	0.442 ***
Itch	0.321 ***	0.313 ***	−0.278 **	−0.389 ***	0.359 ***
Sleep	0.456 ***	0.409 ***	−0.435 ***	−0.488 ***	0.467 ***
Social life	0.331 ***	0.220 *	−0.228 *	−0.324 ***	0.383 ***
Work and other daily activities	0.265 ***	0.132	−0.345 ***	−0.349 ***	0.246 **
Peace of mind	0.399 ***	0.333 ***	−0.390 ***	−0.520 ***	0.405 ***
Sexual life	0.241 *	0.197 *	−0.245 **	−0.320 ***	0.356 ***
Shame	0.331 ***	0.289 ***	−0.237 *	−0.346 ***	0.407 ***
Skin involvement	0.288 ***	0.315 ***	−0.224 *	−0.290 **	0.341 ***

* *p* < 0.05; ** *p* < 0.01; *** *p* < 0.001. BMI: body mass index; DoPI: duration of psoriatic illness; GSI: Global Severity Index; HAM-A: Hamilton Anxiety Rating Scale; HAM-D: Hamilton Rating Scale for Depression; MCS: SF-36 mental component summary; PCS: SF-36 physical component summary; SF-36: Short Form Health Survey 36; SCL-90: Symptom Checklist.

**Table 4 brainsci-14-00865-t004:** Regression analysis (N = 112). Sociodemographic and clinical characteristics of the sample and Psodisk subscale scores are used as independent variables; psychopathological scales are the dependent variables.

					SF-36				SCL-90	
	HAM-A B (95% IC)		HAM-D B (95% IC)		PCS B (95% IC)		MCS B (95% IC)		GSI B (95% IC)	
Adjusted R^2^	0.440		0.300		0.382		0.390		0.310	
Age	0.05	(−0.07/0.17)	−0.06	(−0.17/0.05)	−0.20	(−0.52/0.12)	−0.26	(−0.56/0.03)	−0.27	(−1.03/0.49)
Gender (male)	−3.92 *	(−7.35/−0.49)	−3.42 *	(−6.52/−0.32)	2.31	(−6.65/11.27)	7.69	(−0.65/16.02)	−14.44	(−35.87/6.98)
BMI	0.13	(−0.10/0.36)	−0.02	(−0.23/0.18)	−0.60	(−1.19/0.00)	−0.46	(−1.02/0.09)	0.10	(−1.33/1.53)
Physical activity (sedentary lifestyle)	2.71	(−1.07/6.49)	4.02 *	(0.60/7.43)	−6.34	(−16.21/3.53)	−2.48	(−11.66/6.70)	8.99	(−14.61/32.58)
Personal history of psychiatric disorders (negative)	−4.84	(−10.33/0.65)	−1.47	(−6.43/3.49)	−0.91	(−15.25/13.43)	4.50	(−8.83/17.83)	−20.07	(−54.35/14.21)
DoPI	−0.14 *	(−0.27/−0.02)	−0.09	(−0.20/0.03)	0.11	(−0.22/0.43)	0.27	(−0.03/0.57)	−0.37	(−1.14/0.41)
Arthritis (absent)	−2.22	(−6.07/1.62)	−0.12	(−3.60/3.35)	2.60	(−7.45/12.65)	−0.22	(−9.57/9.13)	−4.18	(−28.21/19.85)
Psodisk										
Health	0.24	(−0.47/0.96)	0.51	(−0.13/1.16)	−1.10	(−2.96/0.76)	−0.47	(−2.19/1.26)	3.10	(−1.34/7.53)
Pain	0.62 *	(0.05 7/1.19)	0.28	(−0.24/0.79)	−2.03 **	(−3.52/−0.54)	−0.46	(−1.85/0.93)	1.52	(−2.05/5.09)
Itch	−0.39	(−0.98/0.19)	−0.18	(−0.71/0.34)	0.49	(−1.04/2.02)	0.21	(−1.21/1.63)	−0.78	(−4.43/2.88)
Sleep	0.68 **	(0.20/1.17)	0.60 **	(0.16/1.04)	−1.46 **	(−2.73/−0.19)	−1.57 *	(−2.75/−0.39)	3.91 *	(0.87/6.95)
Social life	0.21	(−0.40/0.82)	0.13	(−0.42/0.68)	0.14	(−1.45/1.73)	0.15	(−1.33/1.63)	0.91	(−2.89/4.72)
Work and other daily activities	−0.29	(−0.82/0.23)	−0.31	(−0.78/0.16)	−0.10	(−1.46/1.26)	−0.31	(−1.58/0.95)	−1.63	(−4.89/1.63)
Peace of mind	0.54	(−0.12/1.20)	0.15	(−0.45/0.75)	−1.07	(−2.80/0.66)	−2.30 **	(−3.91/−0.69)	0.91	(−3.22/5.05)
Sexual life	−0.15	(−0.77/0.48)	−0.30	(−0.86/0.27)	−0.22	(−1.85/1.41)	−0.05	(−1.57/1.46)	1.23	(−2.67/5.14)
Shame	0.13	(−0.47/0.72)	0.12	(−0.42/0.66)	0.12	(−1.43/1.67)	0.10	(−1.34/1.54)	1.56	(−2.15/5.26)
Skin involvement	−0.02	(−0.75/0.72)	0.04	(−0.63/0.71)	1.33	(−0.59/3.26)	0.89	(−0.90/2.68)	−0.22	(−4.83/4.38)

* *p* < 0.05; ** *p* < 0.01. BMI: body mass index; DoPI: duration of psoriatic illness; GSI: Global Severity Index; HAM-A: Hamilton Anxiety Rating Scale; HAM-D: Hamilton Rating Scale for Depression; MCS: SF-36 mental component summary; PCS: SF-36 physical component summary; SF-36: Short Form Health Survey 36; SCL-90: Symptom Checklist.

## Data Availability

The data that support the findings of this study are available from the corresponding author, S.C., upon reasonable request. The data are not publicly available due to privacy and ethical restrictions.

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
