# Peer review of "Factors Associated with Anxiety, Depression, and Quality of Life in Patients with Psoriasis: A Cross-Sectional Study"

_brainsci, 2024, doi:10.3390/brainsci14090865_

Round 1

Reviewer 1 Report

Comments and Suggestions for Authors

This observational study explores the interplay between mental and physical health in patients with psoriasis, which is an important and clinically helpful topic.  Please refer to the following observations:

Abstract: Consider using patient-first terminology, i.e., „patients with psoriasis” or „patients diagnosed with psoriasis.” The same recommendation applies to the body text.

Were all the instruments used to assess psychiatric variables validated in the Italian population? The same question applies to SF-36.

Any inclusion criteria about the type of current treatment for psoriasis? Also, there is no inclusion of the treatment as a moderator between psoriasis and depressive/anxiety/QoL. It is well known the fact that corticosteroids, methotrexate, cyclosporine, and even biological therapies may have an impact on mental health, as specified in each of the drugs’ summary of product characteristics. Therefore, the possibility of affective/anxiety disorders due to a pharmacological agent can not be excluded in this population.

Exclusion criteria: What about other comorbid organic diseases that may severely impact the monitored parameters, like depression/anxiety/QoL? Were those conditions listed in a table („Clinical characteristics of the sample”) stabilized at the study visit? Were the patients screened for other significant stressors that may have appeared recently in their lives, except for psoriasis-related worries? Were substance abuse and substance dependence excluded?  Of course, this is an observational study, so no causal relationship can be determined, but even so, controlling for the most relevant variables that may moderate the relationship between somatic and psychiatric health in a population is needed, since this is the main hypothesis of the study.

Were the 97.3% of the subjects who drank alcohol regularly meeting the criteria for alcohol use dependence?

The current HAMA and HAMD mean scores may certainly be influenced by the ongoing treatment with antidepressants and/or anxiolytics in the 10.7% + 8% of the subjects. This should be mentioned in the „Limitation” section.

Consider splitting the text on page 6 into multiple paragraphs, because it is hard to follow.

All the tables and figures require numbers and need to be referenced in the text accordingly. Their titles should also be a little more detailed.

Page 9-10- The paragraph about pro-inflammatory cytokines may be related to the current biological therapies interfering with the bioactivity of such cytokines (e.g., IL-12, IL-33). Based on this aspect, a discussion about the subgroup of subjects who may eventually have been treated with monoclonal antibodies would have been interesting.
Page 11- „„Peace of mind” SF-36 subscale” is probably an error, since this is a Psodisk item.

Reviewer 2 Report

Comments and Suggestions for Authors

These are my comments to the authors

The objective, according to the study was: “This study aimed to identify factors influencing anxiety, depression, and quality of life (QoL) in psoriasis patients”, therefore, the title should be adjusted accordingly, I suggest to write something like: Factors associated with anxiety, depression, and quality of life (QoL) in psoriasis patients: A cross-sectional study”

The abstract lack proper results, i.e. there are no numbers of results and their significance, no OR, CI and p values

The clinical assessment under the methodology should be headed according to each scale, also the scales should be identified as covariates or outcome variable, the scales has to be validated mentioning the Cronbach alpha for the Italian version

The psoriasis diagnosis has to follow a well-known criteria with a reference

The heading subjects should be replaced with study design and settings, also, the term This is an observational naturalistic study could be replaced with a better description

Since the objective was to identify factors influencing anxiety, depression, and quality of life (QoL) in psoriasis patients, what is the rationale for the t-test between those with family history vs those without, was this in the objectives? and correlation analysis the same! The correlation study is not new, instead the objective investigated the association with goes in line with regression analysis!

The regression analysis has to be properly mentioned in details, was it a backward step wise model? Was this preceded with a univariate model, was it a linear or binary model, why did the authors did not use cut-off values for severe anxiety, depression, etc.?

Also, in general, the text describes results and discussion only for the results of the multivariable analysis, the univariate analysis results are only presented in the tables

Round 2

Reviewer 1 Report

Comments and Suggestions for Authors

Thank you for taking into account my suggestions.

Note that "Assessment tools" would be a more appropriate title for section 2.2.

Regarding the motivation in lines 422-4234, I would recommend erasing it or rephrasing it because, in its current form, it is ambiguous (since no general population, as a control group, was included in this study).

Reviewer 2 Report

Comments and Suggestions for Authors

The authors amended and improved the quality of the manuscript

Author Response

We would like to express our further gratitude to the referee for their valuable contribution in improving our manuscript.